# Structural Optimization of Self-Supporting Rectangular Converging-Diverging Tube Heat Exchanger

**Feng Jiao [1], Ming Wang [1], Meilin Hu [2] and Yongqing He [3,*]**

[1] School of Chemical Engineering, Kunming University of Science and Technology, Kunming 650500, China; jiaofeng0526@163.com (F.J.); wm19980207@163.com (M.W.)

[2] Yunnan PetroChina Kunlun Gas Co., Ltd., Kunming 650000, China; mailhumeilin525@163.com

[3] Chongqing Key Laboratory of Micro-Nano System and Intelligent Sensing, Chongqing Technology and Business University, Chongqing 400067, China

[*] Correspondence: yqhe@ctbu.edu.cn

**Abstract:** A three-dimensional numerical investigation of turbulent heat transfer and fluid flow characteristics of the new heat exchanger and self-support of a rectangular converging-diverging (SS-RCD) tube bundle heat exchanger with different inserts was performed. The values of the Reynolds number varied from 27,900 to 41,900. The baseline case (without an insert) was compared with two enhanced configurations: one circular hole in the baffle plate (one-circle case) and a rectangular hole in the baffle plate (one-rectangle case). Compared with the baseline case, the airside Nusselt number ($Nu$) of the enhanced cases improved by 39.6~48.0% and 36.2~40.2% and had an associated friction factor ($f$) penalty increase of 53.9–66.7% and 60.7–77.8%, respectively. The baseline case was compared with three enhanced configurations: one-circle case, two-circle case, and three-circle case baffle plate. Compared with the baseline case, $Nu$ of the enhanced cases improved by 39.6–48.0%, 36.2–45.4%, and 35.0–44.2%, with f penalty increases of 53.9–66.7%, 44.9–60.0%, and 43.8–60.0%, respectively. The overall performance was conducted by heat transfer enhancement factor ($\eta$). It was found that the one circle case obtained the best overall performance. The numerical results were analyzed from the viewpoint of the field synergy principle. It was found that the reduction in the average intersection angle between the velocity vector and the temperature gradient ($\theta$) was one of the essential factors influencing heat transfer performance.

**Keywords:** self-supporting; converging-diverging tube; baffle plate; heat transfer enhancement; hole; insert

## 1. Introduction

In recent decades, the high-pressure drop caused by the insert in the tube and the enhancement of heat exchange performance, as well as the research of the tube insert in the heat exchanger, has received significant attention. Heat transfer enhancement (HTE) devices can be divided into active and passive technologies. The latter have higher priority because they do not require external resources and can be used in existing shell and tube heat exchangers [1]. In contrast, the shell side is changed due to its huge size, irregular structure, and various baffles being more complicated. Performance of shell and tube heat exchangers under multiple conditions and configurations is studied in literature both experimentally and numerically [2–4]. In the field of shell-and-tube heat exchangers (STHX), there are mainly research directions, such as twisted types [5], porous media [6], surfaces and fins [7], nanofluid [8], vortex generators [9], phase-change materials [10], and so on. Reference [11] proposed a periodic unit pipeline model for the numerical simulation of the baffle rod heat exchanger. The comparison with experimental data and actual heat transfer engineering correlations shows that the model has high accuracy and reliability. In Reference [12], several shell-and-tube heat exchangers with arcuate baffle heat exchangers and spiral baffle heat exchangers with spiral angles of 20°, 30°, 40°, and 50°

were tested and compared. By considering the Nusselt number and drag coefficient, it was found that the heat exchanger with a spiral baffle angle of 40° had the best performance. Reference [13] proposed a new type of spiral heat exchanger structure composed of a circumferentially overlapping three-part spiral baffle shell and a tube heat exchanger to overcome the natural interval fan-shaped baffle in the equilateral triangle tube layout spiral baffle heat exchanger limitations caused by board unevenness.

Inserting baffles can increase the flow resistance while increasing turbulence and significantly improving the overall heat transfer performance. Reference [14] proposed an improved structure of ladder-type fold baffle, and the structure of the trapezoidal baffle was optimized. Numerical results show that the shell-side heat transfer coefficient of the enhanced heat exchanger increased by 82.8–86.1%. Reference [15] found that, under the same pump power condition, the overall performance ($h/\Delta$P) of the two-layer spiral baffle as 10% higher than that of the single spiral baffle. Reference [16] proposed a continuous spiral baffle. The heat transfer coefficient and pressure drop of the new STHX were compared with the heat transfer coefficient and pressure drop of STHX with segmented separators. The results show that compared with the heat transfer plate, the heat transfer coefficient increased by nearly 10% using the continuous spiral baffle. Reference [17] proposed a novel design of a tube structure with a cosine wave. The results show that as the flow of hot water increased, the coefficient of thermal performance decreased. It was found that the thermal performance factor of the wave tube was greater than the thermal performance factor of the smooth tube. A new type of hexagon clamping anti-vibration baffle shell and tube heat exchanger (HCB-STHX) was proposed to overcome the vibration vulnerability of round rod baffle shell and tube heat exchanger (RRB-STHX) [18]. It was found that compared, with the parallel arrangement of the baffles, the vertical arrangement of the baffles can enhance heat transfer. Still, it is not convenient to manufacture and assemble. In addition to the above, there are staggered baffles [19], louver baffles [20], three-lobed hole baffles [21], and rod-shaped baffles [22].

It can be seen from the above summary that there are many researches on STHE. However, the usage of tube metal material largely results from the thick heat transfer tube wall (2–3.5 mm). Compared with the shell-and-tube heat exchanger, the plate metal as a heat transfer element is thinner (0.5–1 mm). Unfortunately, the plate heat exchanger has a poor leak-proof quality. Integration of the advantages of these two kinds of heat exchangers and the rectangle tube bundle heat exchangers was presented by Deng Xianhe [23]. Although the rectangle tube bundle heat exchangers had good heat transfer performance [24], the experimental results showed that the plate of rectangle tube creates deformation when the pressure increases. Thus, we [25] presented the self-support of converging-diverging tube (SS-RCD) bundle heat exchanger to overcome this limitation. The structural scheme of self-support tube is shown in Figure 1. The rectangle tube becae a self-support tube by pressing the tube and could resist deformation. By inserting three different inserts in the channel, it was found that the baffle plate (BP) case obtained the best overall performance, followed by the twist tape case and regularly spaced twisted-tape (RSTT) case, while the baseline case was the worst [26].

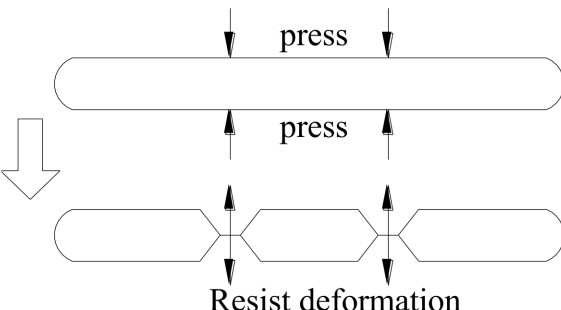

**Figure 1.** Structural scheme of the self-support tube.

Based on the existing insert, thebaffle plate, the influence of hole shape and number of holes on flow characteristics and heat transfer performance of heat exchanger were studied. In order to further improve the heat transfer performance of the heat exchanger, the influence of the circular hole and rectangular hole was considered. The Reynolds numbers varied from 27,900 to 41,900. This research can offer technical references for the design and construction of a high efficiency heat exchanger.

## 2. Mathematical Model

### 2.1. Physical Model

Figure 2 presents the cross section view of the shell side of a SS-RCD tube bundle heat exchanger. The tube in the central region is the smooth tube, and the tube on the sides is the converging-diverging tube (the size is shown in Figure 2b). The calculation element can be defined as the shadow part of the figure, as the construction has the symmetry characters in the z-direction. The sizing used in drawing is the millimetric system. Figure 3 shows the structure of the baffle plate and its insert position. The baffle plate is inserted in the central region of the channel ($l = 12$ mm).

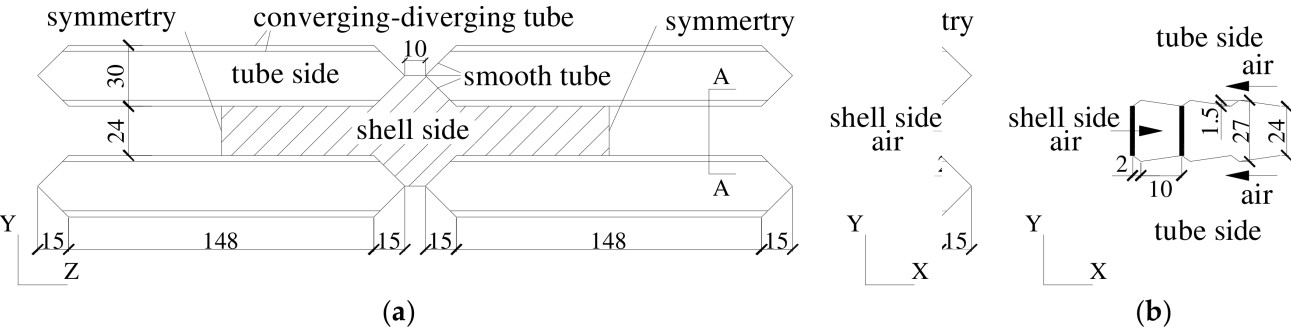

**Figure 2.** Cross section view of the shell side of the SS-RCD tube bundle heat exchanger. (**a**) Schematic diagram of heat exchanger. (**b**) Cross section of A-A.

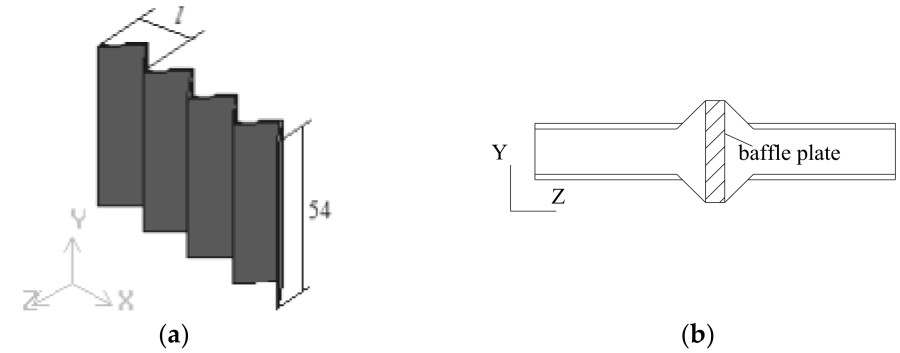

**Figure 3.** Structure of the baffle plate and its insert position (units: mm). (**a**) Structure of the baffle plate. (**b**) Position of the baffle plate.

From the above description, we can see that both the converging-diverging tube bundle and the inserts have periodic geometry characteristics in the x-direction. Thus, the periodic boundary type can be applied in the air flow direction.

The schematic diagram of the structure after the opening of the baffle plate is shown in Figure 4. The opening position was in the central area of the baffle plate, and the diameter of the round hole and the side length of the rectangular hole were both 6 mm. The thickness of the baffle plate was ignored in the calculation. Table 1 shows the parameters of numerical simulation cases.

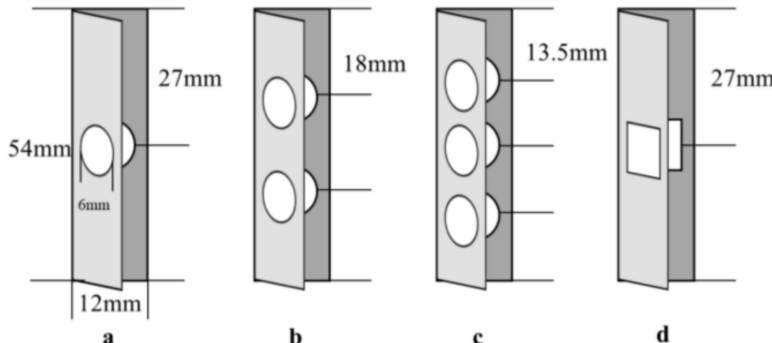

**Figure 4.** Schematic of the structure of the perforated baffle plate. (**a**) One-circle case; (**b**) two-circle case; (**c**) three-circle case, (**d**) one rectangle case.

**Table 1.** The parameters of the numerical simulation cases.

| Case | Hole Shape | Number of Digging Holes | Digging Hole Size |
|---|---|---|---|
| Baseline case | Without insert | - | - |
| BP case | Baffle plate without a hole | 0 | - |
| 1-Circle case | Round | 1 | R = 6 cm |
| 2-Circle case | Round | 2 | R = 6 cm |
| 3-Circle case | Round | 3 | R = 6 cm |
| 1 Rectangle case | Square | 1 | L = 6 cm |

### 2.2. Governing Equations and Boundary Conditions

The air was considered incompressible with constant properties, and the pipe wall adopted the normal wall temperature boundary condition, while the wall temperature was $T_w$ = 343 K. The buoyancy and radiation effects were neglected. Periodic boundary type was applied in the air flow direction. The other boundary conditions are shown in Figure 5. Due to the high thermal conductivity of the tube wall, the conductivity of the wall was ignored.

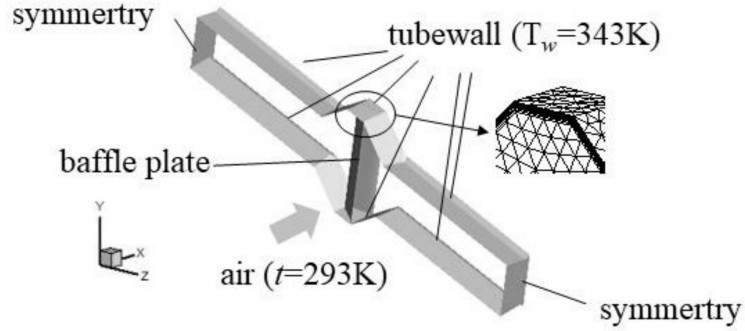

**Figure 5.** Schematic diagram of boundary conditions and enlarged view of the local grid.

For liquids, the continuum model is applicable, so the Navier-Stokes equation based on the continuum assumption can be used to describe it. The governing equations of incompressible steady-state flow in a channel are as follows:

Continuity equation:

$$\frac{\partial(\rho u_i)}{\partial x_i} = 0, \tag{1}$$

Momentum equation:

$$\frac{\partial(\rho u_i u_j)}{\partial x_j} = -\frac{\partial p}{\partial x_i} + \frac{\partial}{\partial x_j}\left[\mu\left(\frac{\partial u_i}{\partial x_j} + \frac{\partial u_j}{\partial x_i}\right)\right] \tag{2}$$

Energy equation:

$$\frac{\partial}{\partial x_j}(\rho u_j C_P T - k\frac{\partial T}{\partial x_J}) = 0 \tag{3}$$

### 2.3. Solution Procedure

The RNG $k$–$\varepsilon$ model of FLUENT was adopted. The coupling of pressure and velocity was implemented by the SIMPLEC algorithm. Segregated manner was selected as the solver type. The standard pressure and the second order upwind discretization scheme for momentum, energy, turbulent kinetic energy, and dissipation energy were employed in the model. Furthermore, the convergence criterion of $10^{-6}$ was chosen for all calculated parameters, except for the energy, where a value of $10^{-7}$ was used.

The RNG $\kappa$–$\varepsilon$ turbulence model was derived from the transient N-S equation using a mathematical method called the renormalization group (RNG). The result of constant analysis in the model was different from that of the standard $\kappa$–$\varepsilon$ model, and a term was added to the $\kappa$ and $\varepsilon$ transport equations. In the RNG $\kappa$–$\varepsilon$ two-equation model, the transmission equations of turbulent kinetic energy $\kappa$ and dissipation $\varepsilon$ were:

$$\frac{\partial}{\partial t}(\rho k) + \frac{\partial}{\partial x_i}(\rho k u_i) = -\frac{\partial}{\partial x_j}(\alpha_k \mu_{eff}\frac{\partial k}{\partial x_j}) + G_K - \rho\varepsilon + S_k \tag{4}$$

$$\frac{\partial}{\partial t}(\rho\varepsilon) + \frac{\partial}{\partial x_i}(\rho\varepsilon u_i) = -\frac{\partial}{\partial x_j}(\alpha_\varepsilon \mu_{eff}\frac{\partial\varepsilon}{\partial x_j}) + C_{1\varepsilon}\frac{\varepsilon}{k}G_k - C_{2\varepsilon}\rho\frac{\varepsilon^2}{k} + S_\varepsilon \tag{5}$$

Among them, $C_{1\varepsilon} = 1.42$, $C_{2\varepsilon} = 1.68$, $\alpha_\varepsilon = \alpha_k \approx 1.393$. At the high Reynolds number, $\mu_{eff} = \mu_i = \rho C_\mu \frac{k^2}{\varepsilon}$, $C_\mu = 0.0845$.

### 2.4. Numerical Methods

The fluid flow and heat transfer simulation used the widely used fluid mechanics commercial software Fluent 15. 0 for simulation, and the ANSYS post-processing software CFD-POST was used for the result analysis. The Cartesian coordinate system was used in the simulation, which is convenient for subsequent calculation and processing. Airside heat transfer coefficient:

$$h = \frac{Q}{A \cdot \Delta T} \tag{6}$$

Among them, the heat transfer:

$$Q = mc_p(T_{out} - T_{in}) \tag{7}$$

Log mean temperature difference:

$$\Delta T = \frac{(T_w - T_{in}) - (T_w - T_{out})}{\ln[(T_w - T_{in})/(T_w - T_{out})]} \tag{8}$$

The Reynolds number, Nusselt number, and fraction factor on the airside [27] were defined as:

$$Re = \frac{d_e u_{max}}{\mu} \tag{9}$$

$$Nu = \frac{h \cdot d_e}{\lambda} \tag{10}$$

$$f = \frac{2 \cdot d_e \cdot \Delta p}{l \rho u_{\max}^2} \qquad (11)$$

The enhanced heat transfer performance was evaluated by performance evaluation criteria (PEC) [28]:

$$\eta = \frac{Nu/Nu_b}{(f/f_b)^{\frac{1}{3}}} \qquad (12)$$

In the formula, $Nu$ and $f$ are, respectively, the Nusselt number and friction factor of the heat exchanger after inserting the insert; $Nu_b$ and $f_b$ are the Nusselt number and friction factor of the empty tube (Baseline Case), respectively. When $\eta > 1$, it indicates that the heat transfer can be enhanced after inserting the insert in the channel; on the contrary, it also indicates that the heat transfer performance is better when the tube is empty.

## 3. Results and Discussion

### 3.1. Meshing and Simulation Verification

The model uses a tetrahedral mesh. The schematic diagram of the local grid is shown in Figure 5. Because the temperature near the wall changed drastically, the mesh near the wall of the model was encrypted, and the dimensionless distance $y+$ from the near-wall mesh node to the wall was controlled below 5. Before the simulation, each model was tested for grid independence. When the error of the result obtained under different grid numbers was less than 2%, the result was considered desirable. Figure 3 shows the test results of grid independence when there was no insert (baseline case) under the Reynolds number of 27,900. The grid numbers were 36,928, 65,150, 100,716, 181,365, 275,936, and 377,660, respectively.

It can be concluded from Figure 6 that the deviation between the drag coefficient and $Nu$ number under the fourth group of grid numbers and the value obtained under the fifth group of grid numbers was within 1%. Considering the computing power of the computer, the fourth mesh size was used to discretize the computational domains for the baseline case. Similar validations were also conducted for other cases.

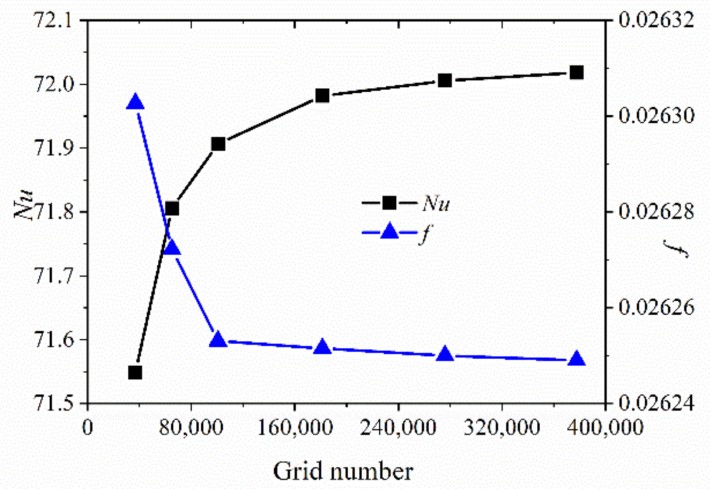

**Figure 6.** Grid independence test.

To further verify the accuracy of the above numerical simulation calculation method, taking the baseline case as an example, we calculated according to the above simulation method and compared the simulation results with the experimental results in [29]. The comparison result is shown in Figure 7. The $Nu$ value error of the two methods was less than 10%, and the resistance coefficient error of the two methods was less than 5%, indicating that the simulation method is feasible.

### 3.2. The Influence of Opening Shape on Heat Transfer and Flow Resistance Performance

To study the influence of the shape of the hole on the heat transfer performance, a round hole and a rectangular hole on the baffle plate were studied separately, and the results are as follows.

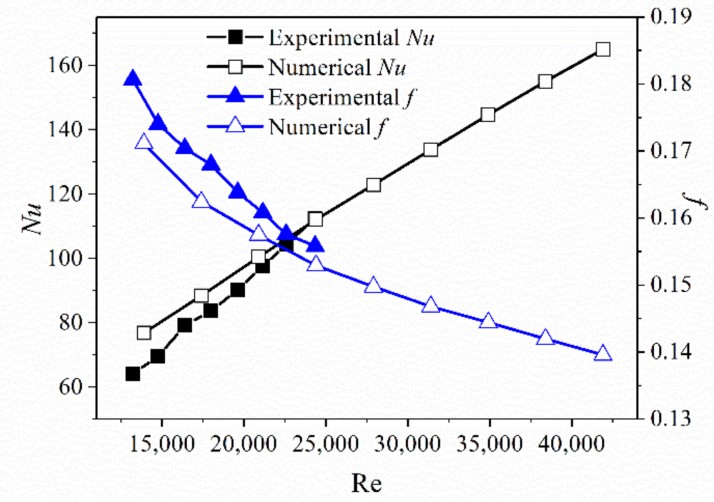

**Figure 7.** Simulation results and experimental results verification.

### 3.2.1. The Influence of the Opening Shape on Heat Transfer and Flow Resistance Performance

Figure 8 shows a velocity distribution and streamline diagram of a section taken at the center of the opening paralleled to the X-Z plane. It can be seen from the figure that after the hole was opened, the velocity of the fluid near the baffle plate obviously increased compared to that when the hole was not opened, and the area of the recirculation area was reduced. This indicates that the fluid disturbance is reduced after the hole is opened and the resistance in the channel is reduced. The flow resistance in the channel can be reduced after opening the hole on the flap.

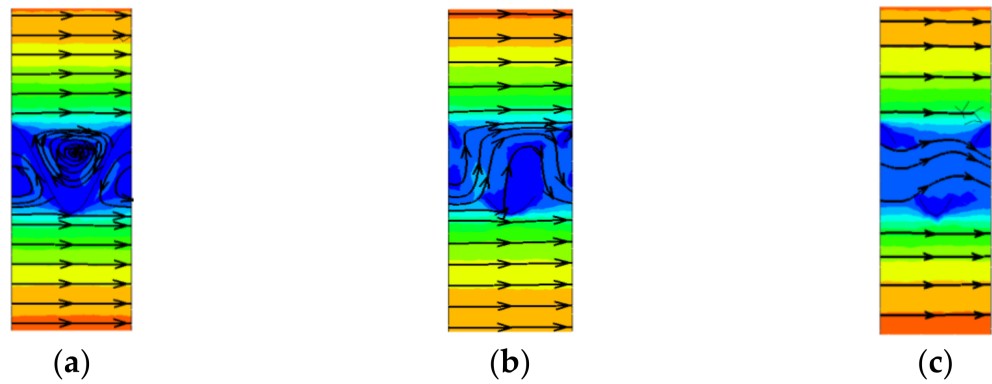

**Figure 8.** Velocity distributions and streamlines on the middle cross section of the hole at Re = 27,900. (**a**) BP case; (**b**) one-circle case; (**c**) one rectangle case.

Figure 9 shows a graph depicting the change of resistance coefficient with Re for empty tubes and different shapes of holes on the baffle plate. It can be seen from the figure that, no matter which structure of the baffle plate was inserted, the resistance increased compared to that without insertion, but the resistance after opening was lower than that without opening and the resistance of a round hole was significantly lower than that of a rectangular hole.

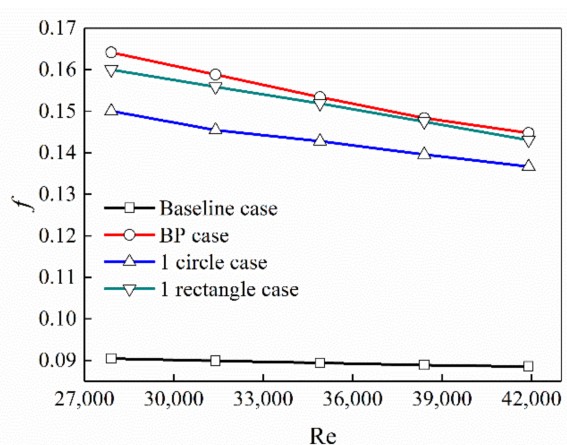

**Figure 9.** The relationship between the shell side drag coefficient and Re.

### 3.2.2. Analysis of Temperature Distribution and Heat Transfer Performance in the Channel

Figure 10 is the temperature distribution on the central section of the X-axis. It can be seen from the figure that the temperature of the fluid in the central area after the opening of the circular hole on the baffle plate was higher than that when the baffle plate was not opened (the area at the minimum temperature decreased) and that the heat transfer in the center area improved after opening the hole. After opening a rectangular hole on the baffle plate, the temperature difference between the fluid and the wall (343 K) in the two sides of the channel significantly reduced, indicating that the heat transfer performance on both sides was weakened and the heat transfer was concentrated in the central area of the channel, while the central area occupied a smaller area of the entire channel, which did not help much to improve the heat transfer performance of the entire channel. Therefore, it can be concluded that the heat transfer performance of the channel after the rectangular hole was opened was not improved but weakened compared to when the non-opened baffle plate was inserted. It can also be seen from Figure 11 that the *Nu* number of the empty tube and the baffle plate changed with Re after opening the hole. The *Nu* number was the lowest when the tube was empty. That is, the heat transfer performance of heat exchange was the worst. After inserting the unperforated baffle plate and the hole in the channel, the heat transfer performance improved, but the heat transfer performance was the worst when the rectangular hole was opened.

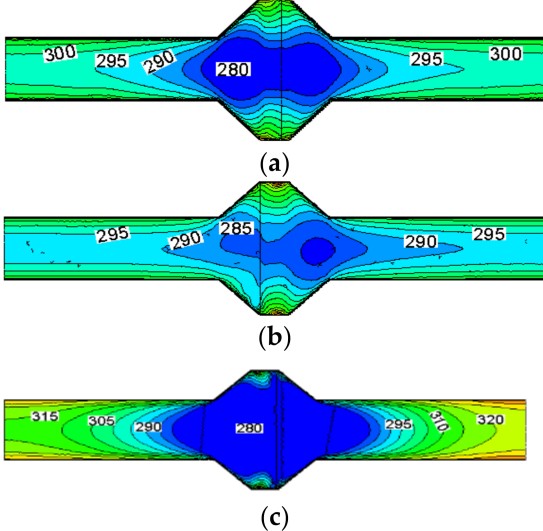

**Figure 10.** Temperature distribution at the center section of the X-axis under three conditions when Re = 27,900. (**a**) BP case; (**b**) one-circle case; (**c**) one rectangle case.

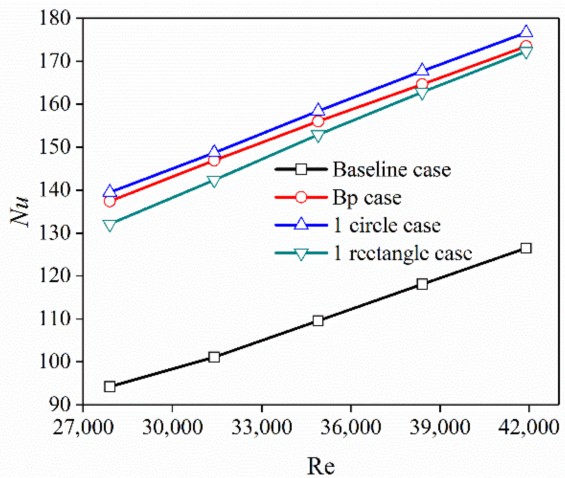

**Figure 11.** Variation of *Nu* on the shell side with Re.

### 3.2.3. Analysis of Field Synergy Theory

Guo et al. [30] proposed that the performance of convective heat transfer not only depends on the velocity and physical properties of the fluid and the temperature difference between the fluid and the wall but also on the degree of coordination between the fluid velocity field and the fluid heat flow field. The better the synergy between the speed and the temperature gradient, the stronger the heat transfer under the same other conditions; and the better the synergy between the two vector fields of the speed field and the temperature gradient means that the angle between the speed and the temperature gradient should be as small as possible and the two should be as parallel as possible [31], that is, the greater the absolute value of the cosine of the angle between the velocity field and the temperature field. The field coordination angle is defined as follows:

$$\theta = \cos^{-1}\left(\frac{u\frac{\partial T}{\partial x} + v\frac{\partial T}{\partial y} + w\frac{\partial T}{\partial z}}{|U||\nabla T|}\right) \tag{13}$$

In the above equation, $u$, $v$, and $w$ are the velocity components in the x, y, and z direction, respectively; $\partial T/\partial x$, $\partial T/\partial y$, and $\partial T/\partial z$ are the temperature gradient components in the x, y, and z direction; $U$ and $\nabla T$ are the velocity vector and the temperature gradient vector.

In this paper, the average volume value of the synergy angle of the various models are compared, and the volume average of the field synergy angle is defined as follows:

$$\theta_m = \frac{\sum_{i,j,k}\theta_{i,j,k}dV_{i,j,k}}{\sum_{i,j,k}dV_{i,j,k}} \tag{14}$$

In the above equation, $\theta_{i,j,k}$ is the field coordination angle values of each volume grid and $V_{i,j,k}$ is each volume unit in the calculation area.

Figure 12 is the volume average of the angle between the velocity vector and the temperature gradient vector in the channel. The fluid in the channel was heated, and the smaller the angle, the better the heat transfer performance. It can be seen from the figure that the angle between the velocity vector of the fluid in the channel and the temperature gradient vector was the largest when the converging-diverging tube was the baseline case. At this time, the heat exchange effect was the worst. After inserting the perforated baffle plate, the coordination angle was significantly reduced, showing that heat transfer was improved. The synergy angle when opening a circular hole was the smallest, and the heat transfer performance was the best.

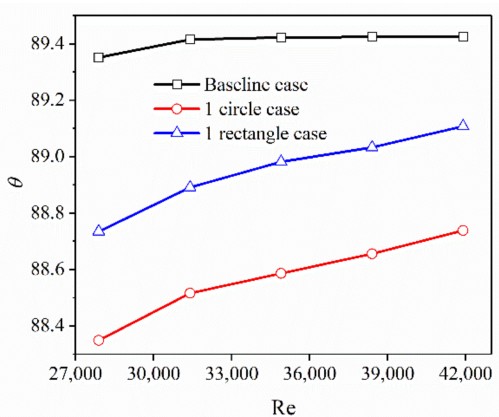

**Figure 12.** Synergy angle distribution in three cases.

*3.3. The Influence of the Number of Holes on the Heat Transfer and Flow Resistance Performance*

3.3.1. Velocity Distribution and Flow Resistance in the Channel

From the above analysis, it can be seen that opening holes are beneficial to reduce the resistance of the fluid in the channel during the passage. Therefore, the influence of the number of openings on the fluid flow in the passage will be further analyzed.

Figure 13 is the velocity distribution and streamline diagram of the section taken at the center of the hole paralleled to the X-Z plane. When there are no opening holes, the central section of the Z-axis was taken as a description section, and only the cross-section of one of the holes was taken as an illustration when there were two holes. It can be seen from the velocity distribution in the figure that the velocity of the fluid near the baffle plate significantly increased after opening holes on the baffle plate. It can be seen from the streamline diagram that the backflow of the fluid on both sides of the baffle plate was obvious when there were no opening holes on the baffle plate, and the area of the backflow vortex was large. However, after opening holes on the baffle plate, the backflow area on both sides of the baffle plate significantly decreased, the area of backflow vortex decreased, and the fluid disturbance decreased, so the resistance of fluid in the channel decreased, showing that the resistance in the channel can be decreased after opening holes on the baffle plate.

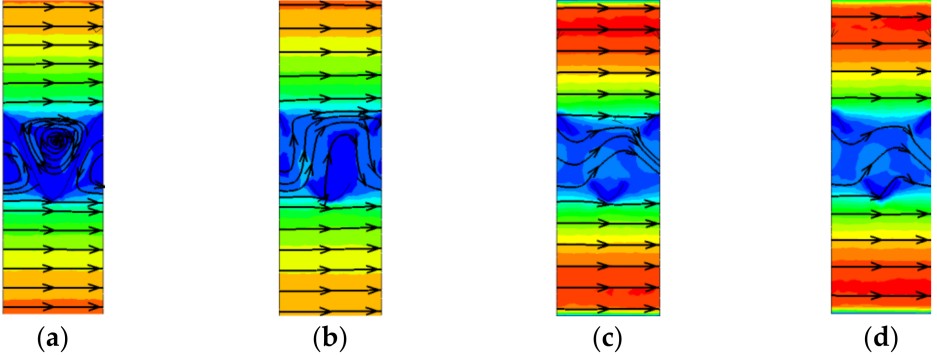

**Figure 13.** Velocity and streamline distributions on the middle cross section of the hole in four cases at Re = 27,900. (**a**) BP case; (**b**) one-circle case; (**c**) two circles case; (**d**) three circles case.

Figure 14 is a diagram showing the change of resistance coefficient with Re for the empty tube of the zoom tube and the different number of holes on the baffle plate. It can be seen from the figure that, no matter which structure of the baffle plate was inserted, the resistance increased compared to the case without insertion, but the resistance in the channel gradually decreased with the number of holes, which is consistent with the conclusions obtained above.

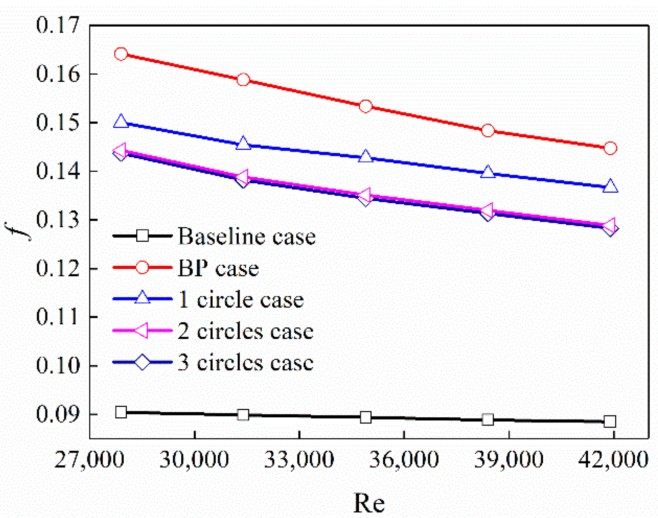

**Figure 14.** Variation of friction factor with Re.

3.3.2. Temperature Distribution and Heat Transfer Performance Analysis in the Channel

Figure 15 shows the temperature of the cross-section taken along the center position of the X-axis. It can be observed from the figure that changing the number of openings on the baffle plate had little effect on the heat exchange effect and temperature distribution on the shell side of the heat exchanger. When the number of openings of the baffle plate changed, the temperature distribution in the fluid center area was higher than that when no holes were opened and the area of the minimum temperature area was reduced. The reason for this phenomenon is: with the upper opening, the fluid can flow away from the opening, but when the opening is not opened, the fluid cannot flow through the central area where the opening is located. Therefore, after opening the hole, the heat transfer in the central area is improved compared to that without opening. Therefore, the temperature in the central area is increased compared to when the hole is not opened.

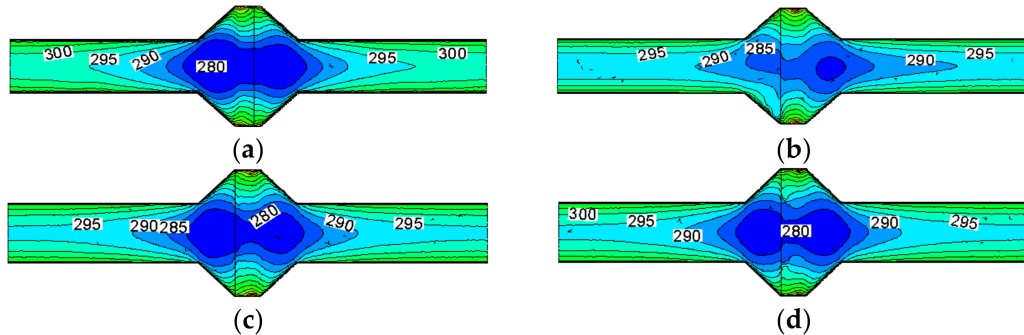

**Figure 15.** Temperature distributions at the X-axis central section in five cases at Re = 27,900. (**a**) BP case; (**b**) one-circle case; (**c**) two-circle case; (**d**) three-circle case.

Figure 16 shows a cross-section taken along the center of the Z-axis to obtain a temperature distribution diagram. A situation like that of Figure 12 can be observed. After opening the holes on the baffle plate, the temperature of the fluid in the central area increased to a certain extent compared with the opening, and the temperature distribution was more uniform than when there was no opening. However, it can be seen from the above two figures that the temperature in the channel did not change very much. Therefore, it can be inferred that changing the structure of the baffle plate does not significantly improve the heat transfer performance of the channel.

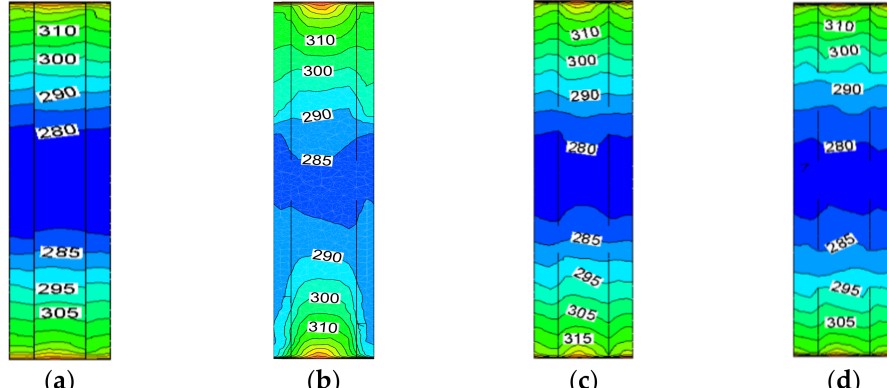

**Figure 16.** Temperature distributions at the Z-axis central section in five cases at Re = 27,900. (**a**) BP case; (**b**) one-circle case; (**c**) two-circle case; (**d**) three-circle case.

Figure 17 shows a variation diagram of the Nusselt number with Re, showing that the converging-diverging tube baseline case and the different structural baffle plates were inserted in the channel and, as can be seen from the figure, that the Nusselt number value was the lowest in the converging-diverging tube baseline case, which indicates that the heat transfer performance in shell side was the worst. After the baffle plate was inserted into the channel, the *Nu* number was greatly improved compared to the case without the baffle plate, which means that inserting the baffle plate on the shell side significantly improved the heat transfer performance. The uniformity of the temperature distribution was also the best when a hole was opened on the baffle plate, which shows that the heat transfer performance of the channel was closely related to the uniformity of the internal temperature distribution.

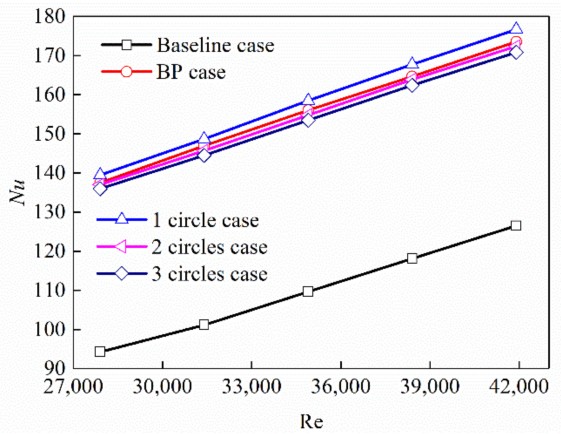

**Figure 17.** Variation of *Nu* with Re in the shell side.

### 3.3.3. Field Synergy Theory Analysis

Figure 18 shows the distribution diagram of the average volume value of the synergy angle in the channel. The maximum angle between the velocity field and the temperature field is the case where the converging tube is an empty tube, and the heat transfer performance was the worst. When the baffle plate was inserted, the included angle became significantly smaller, indicating that the inserting of the baffle plate enhances the heat transfer performance [25]. The different numbers of holes in the baffle plate were compared, and the synergy angle when a hole was opened in the center of the baffle plate was the smallest. According to the analysis of field synergy theory, the heat transfer performance on the shell side of the heat exchanger was the best.

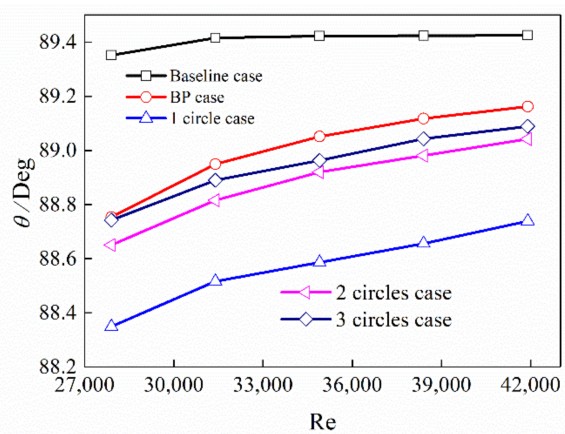

**Figure 18.** Distributions of synergy angle in some cases.

### 3.3.4. The Comprehensive Heat Transfer Performance Analysis

Figure 19 shows a comprehensive factor diagram for several situations. The comprehensive heat transfer factor was above 1, indicating that the insertion of different structured baffle plates in the channel has the effect of enhancing heat transfer. However, the heat transfer performance of the heat exchanger was not significantly improved after the round holes were opened on the double-baffle plate, but the resistance in the channel was significantly reduced. The comprehensive heat transfer performance after the hole was better than that without the hole, which was a significant improvement. It can be seen from the figure that the comprehensive heat transfer factor value was the largest when a circular hole was opened on the baffle plate, indicating that the comprehensive heat transfer performance of the heat exchanger was the best at this time.

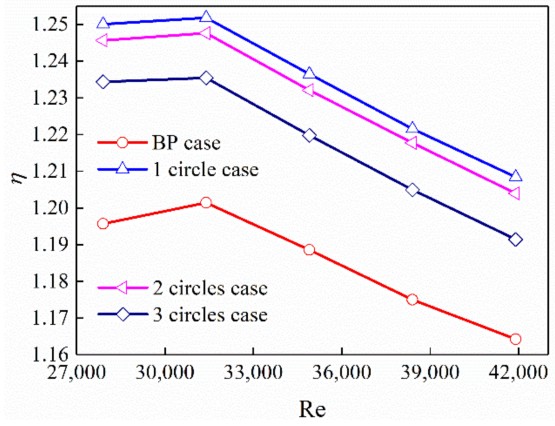

**Figure 19.** Variation of comprehensive heat transfer factor with Re.

### 4. Conclusions

Based on the existing research, this paper studies the opening of the baffle plate inserted in the self-supporting rectangular convergent tube heat exchanger. The influence of the shape of the opening, the number of openings, and the arrangement of the openings on the heat transfer performance are considered.

(1)  The baseline configuration (without insert) was compared with two enhanced configurations (with inserts): a one-circle hole in the baffle plate (one-circle case) and a rectangle hole in the baffle plate (one-rectangle case). Compared with the baseline case, the airside Nusselt number ($Nu$) of the enhanced cases improved by 39.6~48.0% and 36.2~40.2%, with an associated friction factor ($f$) penalty increase 53.9–66.7% and 60.7–77.8%, respectively.

(2) The baseline case was compared with three enhanced configurations: one-circle case, two-circle case, and three-circle case baffle plate. Compared with the baseline case, *Nu* of the enhanced cases improved by 39.6–48.0%,36.2–45.4%, and 35.0–44.2%, with an *f* penalty increase of 53.9–66.7%, 44.9–60.0%, and 43.8–60.0%, respectively. The overall performance was conducted by heat transfer enhancement factor ($\eta$). It was found that the one-circle case obtained the best overall performance.

(3) The self-support of rectangular converging-diverging tube bundle heat exchanger greatly improved the load that the heat exchanger plate can bear, that is, it improved the strength of the heat exchanger. Furthermore, the addition of the insert can improve the heat transfer performance. Therefore, this kind of heat exchanger has certain advantages in industrial application.

**Author Contributions:** Conceptualization, Y.H.; data curation, M.W. and M.H.; numerical simulation, F.J.; funding acquisition, Y.H.; investigation, F.J.; methodology, M.W. and Y.H.; data calculation, M.H.; validation, F.J.; writing—original draft, F.J.; writing—review and editing, F.J., M.W. and Y.H. All authors have read and agreed to the published version of the manuscript.

**Funding:** This research was funded by the National Natural Science Foundation of China (Grant No. 11962010).

**Conflicts of Interest:** The authors declare no conflict of interest.

## Nomenclature

| | | | |
|---|---|---|---|
| $A$ | total heat transfer surface area (m$^2$) | $U$ | velocity vector |
| $C_p$ | specific heat (J/kg K) | $v$ | velocity in y-direction |
| $de$ | hydraulic diameter (m) | $w$ | velocity in z-direction |
| $f$ | friction factor | **Greek symbols** | |
| $h$ | heat transfer coefficient (W/m$^2$ K) | $\eta$ | the thermal enhancement factor |
| $L$ | the length of the computation in x-direction (m) | $\lambda$ | thermal conductivity (W/(m K)) |
| $m$ | mass flow rate (kg/s) | $\mu$ | dynamic viscosity (Pa s) |
| $Nu$ | Nusselt number | $\theta$ | the intersection angle (deg) |
| $\Delta p$ | air-side pressure drop (Pa) | $\rho$ | density (kg/m$^3$) |
| $Q$ | heat transfer capacity (W) | **Subscripts** | |
| Re | Reynolds number | $b$ | baseline case |
| $T$ | temperature (K) | $in$ | inlet parameter |
| $\Delta T$ | log mean temperature difference (K) | $m$ | mean value |
| $\bigtriangledown T$ | temperature gradient | $out$ | outlet parameter |
| $u$ | velocity in x-direction | | |
| $u_{max}$ | velocity at the minimum area | $w$ | tube wall |

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
