# Peer review of "Structural Optimization of Self-Supporting Rectangular Converging-Diverging Tube Heat Exchanger"

_energies, doi:10.3390/en15031133_

Round 1

Reviewer 1 Report

This study is on a numerical simulation of heat transfer and flow characteristics of flows through baffle plate. The effect of the hole shape of the baffle plate had been examined. Although they showed that the performance could be improved by drilling the holes in the baffle plate, the structure and configuration of the target changer are not clearly shown at all. The computed domain is also not clear. Therefore, the manuscript is not acceptable as a journal paper.

1. The author should more explain the targeted heat exchanger using conversing-diverging rectangular tubes. And, the merit of the configuration, problems in the design, and what you want to clarify in the numerical simulation, should be carefully explained. In this manuscript, only two lines at the last of the introduction is on the object.

2. Figure 1 is insufficient to explain the configuration. A diagram showing the overall structure of the heat exchanger is required. The flow direction and tube wall are unclear. The thickness of the baffle plate is also unclear.

Generally, baffle plates in shell and tube heat exchangers are installed to support and keep the tube pitch and to form cross flows in the tube bundle. The relative position of the baffle plates to the tubes should be carefully shown.

3. Figure 2 is insufficient to explain the configuration. It is unclear how the two plates in each figure are placed in the tube bundle.

4. Three-dimensional configuration of the calculation domain should be carefully shown. The boundary conditions at the inlet, outlet, heat transfer wall, and boundaries to sub-channels should be clearly shown. What is the importance of converging-diverging channel shape?

5. In the numerical simulation model, the reason for applying the values of the constants in Eqs. 1 and 2 should be shown.

6. In the consideration of the effect of the grid size, not grid number but grid size is effective for the readers.

7. You explained that the air inlet temperature is set to 293 K, and the inlet and outlet are periodic boundary conditions. However, in Eq. 8 the inlet temperature is set at 303 K, and the inlet velocity is set to the constant at u_in. These things are inconsistent.

8. In the reviewer's understanding, the numerical calculation had been conducted for the shell side flows at the part of the tube bundle, what is the meaning of the assumption of "insulation around". Is there any boundary to the surroundings?

9. In the calculation, the treatment of the heat resistance between the heat transfer wall and baffle plate should be carefully explained. If heat conduction from the heat transfer wall to the baffle plate exists, the thickness and material of the baffle plate affects the fin efficiency.

10. Although the author assumed that the fluid is incompressible and the flow is laminar, you explained that the target fluid is air and the Reynolds number is over 10 power 4th. These things are inconsistent. The explanation is required.

11. The reviewer cannot understand where the area in Figs. 5 and 10 is in the computational domain. Calculated flow field in the area shown in Figs. 7 should be shown.

12. The definition of the comprehensive heat transfer performance should be shown. Is the parameter same with eta in Eq. 16? If so, you should indicate the vertical axis in Fig. 16.

13. In the discussion, you explained only on the calculated results, especially on the performance. You should carefully discuss based on the flow structure.

Reviewer 2 Report

The present work could be reconsidered after addressing the following comments:

  1. The novelty of the work must be clearly addressed and discussed, compare your research with existing research findings and highlight novelty, (compare your work with existing research findings and highlight novelty).
  2. The motivation and contribution of this works must be clarified in the introduction section.
  3. The introduction should contain a final paragraph depicting the structure of the paper and introducing the contents of the following sections.
  4. Please add some real pictures of the experimental setup and equipment.
  5. More explanation is required in the conclusions section.
  6. Conclusion: The future scope of the work should be provided.
  7. Where is the Nomenclature?

Reviewer 3 Report

The paper deals with numerical evaluation of the heat transfer performance for a rectangular convergent heat exchanger.

 Some points need revisions considering the comments addressed to the authors.

  • The Nomenclature section should be introduced in the paper due to the multitude of parameters.
  • Line 48 - it is specified “spiral baffle angle of 40° has the best performance”. Specify why this has the best performance.
  • Line 94 – it is specified “The thickness of the baffle plate is ignored in the calculation” .Why is it neglected?
  • Line 127, replace tw with Tw.
  • Line 228, it is better to write U is s the velocity vector and after that specify grad T is the temperature gradient vector.
  • The Conclusions are limited. The Authors should indicate more clearly which is the practical importance with reference to the experimental results obtained in the literature. It is not clear how these results may be useful in practical applications.

Round 2

Reviewer 2 Report

The authors have addressed my comments satisfactorily. I have no further questions.